# Simplified Genomic Data Revealing the Decline of *Aleuritopteris grevilleoides* Population Accompanied by the Uplift of Dry-Hot Valley in Yunnan, China

**DOI:** 10.3390/plants12071579

**Published:** 2023-04-06

**Authors:** Xue-Ying Wei, Ting Wang, Jin Zhou, Wei-Yue Sun, Dong-Mei Jin, Jian-Ying Xiang, Jian-Wen Shao, Yue-Hong Yan

**Affiliations:** 1College of Life Sciences, Anhui Normal University, Wuhu 241000, China; 2Key Laboratory of National Forestry and Grassland Administration for Orchid Conservation and Utilization, The Orchid Conservation and Research Center of Shenzhen, Shenzhen 518114, China; 3Anhui Key Laboratory of Biological Resources Conservation and Utilization, Anhui Normal University, Wuhu 241000, China; 4Yunnan Academy of Biodiversity, Southwest Forestry University, Kunming 650224, China; 5Eastern China Conservation Centre for Wild Endangered Plant Resources, Shanghai Chenshan Botanical Garden, Shanghai 201602, China

**Keywords:** GBS data, population genetics, divergence time, reconstruct ancestral state

## Abstract

Understanding the evolutionary history of endangered species is crucial for identifying the main reasons for species endangerment in the past and predicting the changing trends and evolutionary directions of their future distribution. In order to study the impact of environmental changes caused by deep valley incision after the uplift of the Qinghai-Tibet Plateau on endangered species, we collected 23 samples belonging to four populations of *Aleuritopteris grevilleoides*, an endangered fern endemic to the dry-hot valleys (DHV) of Yunnan. Single-nucleotide variation sites (SNPs) were obtained by the genotyping-by-sequencing (GBS) method, and approximately 8085 SNP loci were identified. Through the reconstruction and analysis of genetic diversity, population structure, population dynamics, evolution time, and ancestral geographical distribution, combined with geological historical events such as the formation of dry-hot valleys, this study explores the formation history, current situation, reasons for endangerment and scientifically sound measures for the protection of *A. grevilleoides*. In our study, *A. grevilleoides* had low genetic diversity (Obs_Het = 0.16, Exp_Het = 0.32, Pi = 0.33) and a high inbreeding coefficient (Fis = 0.45). The differentiation events were 0.18 Mya, 0.16 Mya, and 0.11 Mya in the *A. grevilleoides* and may have been related to the formation of terraces within the dry-hot valleys. The history of population dynamics results shows that the diversion of the river resulted in a small amount of gene flow between the two clades, accompanied by a rapid increase in the population at 0.8 Mya. After that, the effective population sizes of *A. grevilleoides* began to contract continuously due to topographic changes resulting from the continuous expansion of dry-hot valleys. In conclusion, we found that the environmental changes caused by geological events might be the main reason for the changing population size of *A. grevilleoides*.

## 1. Introduction

Tectonic collisions and climate change are important factors that affect the evolution of species [1]. Geological events have continuously uplifted mountains because of the collision of the Indian plate and the Eurasian plate, such as the formation of the Qinghai-Tibet Plateau in China, which is described as the “cradle of evolution”. This has created rich topography and changeable climatic conditions, providing an important driving force for species differentiation [2,3]. For example, for *Quasipaa boulengeri* Guenter, because of the complex orogenesis, its genetic structure differentiated along geographical units [4]. Wu et al. proved that the two increases in the diversification rate of *Oreocnide* Miq. plants were affected by climate change [5].

The Tibet Plateau is an important natural landscape with a special ecosystem and has the highest species diversity in the northern hemisphere. At least 12,000 species of vascular plants are found on the Chinese part (Yunnan, Xizhang, Sichuan provinces, and so on) of the Tibet Plateau; these species belong to over 1500 genera, and approximately 20% of the species are endemic [6,7]. Due to the uplift of the plateau and the descent of the river valley, the special landform of “dry-hot valley” was born [8,9]. Dry-hot valleys have peculiar climates. Their formation depends on a complex geographical environment and the resulting local microclimates [10]. When the water vapor condenses in the valley, it causes heat release, reduces the water availability, and increases the air temperature, forming a warm and dry environment. This environment became a refuge for ancient plants and their descendants on the ancient southern continent during the Pleistocene period [11,12]. A total of 163 families, 1038 genera, and 3217 species and varieties of native seed plants have been recorded in the dry-hot valleys (DHVs) of southwest China [9], accounting for about a quarter of all vascular plants on the Qinghai-Tibet Plateau.

During the Miocene (23–5.3 Mya), the uplifted terrain and the deep incision of river valleys of Yunnan province changed the distribution of biological and geological zonality. Dry-hot valleys with extreme hydrothermal conditions lead to excessive water loss and make forest vegetation difficult to restore; thus, large areas of land are barren, forming savannah-like vegetation and arid shrubs [8,9,13]. The savanna-like vegetation in Yunnan is morphologically similar to that of the savannas in India and Myanmar, and the geological events in Yunnan may be an important factor affecting the flora and differentiation of the savanna-like vegetation in these dry-hot valleys [9]. This special geological background also created a special biological evolutionary history of local plants.

The study of the evolutionary history of biological groups can help elucidate the reasons for the formation and endangerment of species in special habitats. It can also provide additional sources of evidence to clarify the nature and timing of geological and climatic events in specific regions [14,15,16]. At present, approximately 748 species of wild plants inhabit the dry-hot valleys of the Jinsha River Basin in Yunnan; these species belong to 111 families and 427 genera [11]. Among them, 78 endemic plant species are dominant in various types of vegetation, including *Terminalia franchetii* Gagnep., *Ziziphus yunnanensis* C.K. Schneid., *Vitex negundo* L., and *Andropogon yunnanensis* Hack. [11]. Zhang and Sun analyzed the genetic structure of *T. franchetii* populations in the Jinsha River Basin and concluded that historical drainage capture events led to the modern distribution and related genetic structure patterns [17]. Other researchers have used genetic methods to explore the relationships between species and geographic locations in dry-hot valleys [17,18]. Many studies have investigated the conservation of species’ genetic diversity through simplified genome and population genetics; however, research on the genetic differentiation and conservation of species living in special habitats, such as hot and dry river valleys, is relatively limited.

*Aleuritopteris grevilleoides* (Christ) G. M. Zhang ex X. C. Zhang is a threatened species [19,20] endemic to dry-hot river valleys, and several populations spread to other arid rocky areas in Yunnan. Before it was categorized as a member of *Aleuritopteris* Fee, it was considered an endemic genus in China and known as *Sinopteris* Christensen and Chink [21]. Therefore, this species was included on the national key protected wild plants list in 1999 but removed from the list in 2021 because of changes in its taxonomic position [22,23]. Wang et al. studied the reproduction of *A. grevilleoides* and found that the growing development of gametophytes and young sporophytes are quite different in various soil substrates. Among them, it grows best in the substrate of 1:1 mixed soil of original soil and humus soil [24,25]. Field investigation and prediction analysis of potential distribution suggested that human disturbances, such as hydropower station construction and road expansion, were one of the main reasons for the destruction of *A. grevilleoides* wild populations [26]. Some researchers have proposed that the endangerment of dry-hot valley vegetation is due to the long-term interference of human activities, which have led to land degradation and extensive soil erosion [10,27]. However, others hold a different view, arguing that dry-hot valley vegetation is mainly affected by the uplift of the Qinghai-Tibet Plateau, river diversion, and other geological events caused by climate and environmental changes [9,17].

Recently, with the rapid development of high-throughput sequencing, conservation genomics studies have successfully explained the internal causes of species endangerment and helped to develop protection strategies. In this study, we analyzed genetic diversity and structure, reconstructed demographic history and ancestral states, and estimated the population divergence time of *A. grevilleoides* in a dry-hot valley. A variety of analytical methods were used here to provide robust data and combined with geological historical events to identify the reasons for the sharp decline in the number of species and re-evaluate the status of *A. grevilleoides* reasonably and accurately.

## 2. Results

### 2.1. Genetic Structure

Based on the clustering of 8085 SNP loci after filtering, the CV value of genetic structure analysis was the smallest when K was 4 (Figure 1 and Figure 2A). When the best grouping value (K) of 4 was selected, 26 studied individuals were grouped into four clusters, and *A. grevilleoides* samples were divided into three genotypes (Figure 1). A small amount of introgression was observed between the three genetic groups of *A. grevilleoides* (QH, KM, and SC; Figure 1). The KM population was far away from other populations and had less gene exchange (Figure 1), and the QH group was similar to the KM group. Principal component analysis (PCA) also revealed four genetic groups of studied individuals (Figure 2B): the *A. albofusca* and *A. grevilleoides* were clearly separated and three genetic groups in *A. grevilleoides*, which was consistent with the results of the structural analysis.

The genetic differentiation coefficient (Fst) values among the 26 samples (Table 1) range from 0.031 to 0.684. The Fst values between the *A. grevilleoides* populations were all less than 0.250. The Fst values between the four populations of *A. grevilleoides* and *A. albofusca* range from 0.543 to 0.684, indicating that the two species were well differentiated. The SC and TD groups had an Fst of only 0.031 and had the same genotype. KM group also had a large differentiation from the other two groups. The degree of genotype differentiation between the KM group and the SC group was much lower than that between the QH group and the SC group.

### 2.2. Genetic Diversity

The statistical analysis of the genetic diversity information for the four populations of *A. grevilleoides* (Table 2) revealed that the QH group had the highest number of private alleles. Nucleic acid diversity Pi ranged from 0.159–0.268, and the SC (Pi = 0.251) and TD (Pi = 0.268) groups had higher genetic diversity. The Fis value can be used as an index of the degree of inbreeding among individuals within a population. Table 2 shows that the Fis value was greater than 0 in each population, indicating that inbreeding existed in all populations within the *A. grevilleoides* populations. The number of private alleles in the SC and TD groups was nearly 50% lower than that in the KM group, indicating that the two groups were less differentiated than the KM group.

### 2.3. Divergence Time Estimation and Reconstruction of Ancestral State

RASP was used to simulate the ancestral distribution of *A. grevilleoides*, and the results (Figure 3A) indicated that an event of vicariance may have occurred in node 16 (Figure 3B). *A. grevilleoides* was probably widely distributed in the ABC area after the differentiation at 3.5 Mya. Subsequently, a vicariance event likely occurred in area A at 0.18 Mya. Two events, dispersal and vicariance, may have occurred at node 14 (Figure 3C,D). The original distribution area was BC, and the distribution then spread to D, after B vicariance occurred. The KM group, which had high homozygosity, was more closely related to the TD and SC groups at 0.16 Mya. This is consistent with the results of genetic differentiation. Node 13 shows that the CD area was the original distribution area, and it is highly likely that a geographical separation (vicariance) between C and D has occurred at 0.11 Mya (Figure 3E). The three differentiations of *A. grevilleoides* (Figure 3A) were both related to the formation of the dry-hot valleys.

### 2.4. Historical Fluctuation of Effective Population Size

The demographic history of the *A. grevilleoides* population was inferred on the basis of 23 individuals using Stairway Plot 2 with folded SFSs and masking singletons. The results based on the 95% confidence interval indicate that *A. grevilleoides* was in danger of extinction between 0.002 and 0.004 Kya. According to the standard line (Figure 4), the population size of *A. grevilleoides* fluctuated between 700 and 800 Kya, rapidly decreasing and then increasing rapidly, which may be related to the historical drainage capture events between the Luquan Wudongde and Jinpingzi Canyon in the dry-hot valley approximately 800 Kya. Then, the population decreased in a stepwise manner at 30–40 Kya, 5–6 Kya, 0.8–1 Kya, 0.15–0.2 Kya, 0.04–0.06 Kya, and 0.01–0.015 Kya. Notably, the six sharp declines and one fluctuation observed in the *A. grevilleoides* population, two were directly related to geographic events of dry-hot valleys.

## 3. Material and Methods

### 3.1. Plant Material

*Aleuritopteris grevilleoides* is mainly distributed in the west of Yunnan Province (Binchuan and Dayao) and the north of Sichuan Province (Qingchuan) in China, according to documentation records [30]. During the field investigation, no wild populations of *A. grevilleoides* were found in Sichuan, and only four populations (Figure 5) were found in Dayao County (SC, TD, and QH) and Kunming (KM), Yunnan. Twenty-three individuals from four wild populations (*A. grevilleoides*) and three individuals from the wild population of its closely related species (*Aleuritopteris albofusca* (Baker) Pic.Serm.) were collected. The collection scope covered the distribution areas found in the field of *A. grevilleoides.* The samples were grouped according to regional distribution (Figure 5). The sample materials for sequencing are all from fresh pinnae, and the whole plant was selected for pressing voucher specimens, dried in an oven, and stored in the herbarium of Shanghai Chenshan Botanical Garden (CSH). The detailed sample information is shown in Table 3.

### 3.2. DNA Extraction and Genotyping by Sequencing Library Construction

Total DNA was isolated from the samples using the Plant Genomic DNA Rapid Extraction Kit (Biomed Gene Technology, London, UK). DNA integrity was tested on 1% agarose gel. High-quality DNA was used to construct sequencing libraries for the genotyping by sequencing (GBS) method [31] (Beijing Novogene Co., Ltd., Beijing, China). Mse I and EcoR I were used for enzyme digestion. Adapters were added to both ends of the fragments, and the tag sequences containing the adapters were amplified by polymerase chain reaction (PCR). After mixing the samples, the DNA bands were recovered by electrophoresis, and the purified products were sequenced on the Illumina HiSeq platform.

### 3.3. Determine Single Nucleotide Polymorphisms (SNPs)

The raw reads obtained by machine sequencing were quality-controlled using FastQC 0.11.9 software, sequencing adapters and primer sequences were removed, and bases with a quality value lower than 20 were cut off. The fern09027 sample was selected for Stacks 2.54 [32] clustering and a pseudo-reference sequence was constructed, and the clean reads after filtering the sample were compared with the reference genome using BWA 0.7.17 [33]. Samtools 1.14 software was used to add headers and remove duplicates, and GATK 4.0 was run for variant detection. Individual sample variation sites and genotype information were extracted to generate vcf format files to obtain SNP markers.

The denovo_map.pl script in Stacks v2.54 [32] was used to perform de novo assembly and mutation detection on the GBS data of *A. grevilleoides*. To ensure the reliability of the downstream analysis, two parameters “--max-missing 0.5” and “--mac 3” in Plink 2 [34] were used to filter the SNPs. Further use Plink 2 to filter Hardy–Weinberg equilibrium (HWE) and linkage disequilibrium (LD), and filter according to *p* < 0.001, r2 < 0.001; The remaining high-quality SNPs after filtering were used for population structure.

### 3.4. Genetic Diversity and Structure

The 8085 remaining high-quality SNPs were applied in the sNMF function of R package LEA [35] to analyze the genetic structure of the 26 individuals. The sNMF settings were iterations = 200 and repetitions = 10 with other arguments set to defaults and the best K evaluated by minimal cross-entropy (CE). Principal component analysis (PCA) was performed by Plink 2 [36], and the resulting graph was drawn using the ggplot2 package in R.

The fixation index (Fst), based on genetic polymorphism data, estimates the difference between the average heterozygosity of a population and the average heterozygosity of the entire population. The observed heterozygosity (Ho), expected heterozygosity (He), nucleotide diversity (Pi), and fixation index (Fst) of all populations were calculated with Stacks v2.54 [37].

### 3.5. Divergence Time and Ancestral State

Biallelic SNPs were extracted from the SNP data by BCFtools, and each pair of SNPs was required to be more than 100 bp [38]. Stange used approximately 1000 SNPs to obtain accurate and strongly supported species [39]. Hybrid species cannot be used for the construction of multispecies coalescent models, which adversely affects the accuracy of the phylogenetic tree. After eliminating the hybrids of each population according to the results of the structure analysis, one to two species were selected from each population for tree construction. The python script snapp_prep.rb was used to set the parameters to “-m 1000” and “-l 100000” (https://raw.githubuserco-tent.com/mmatschiner/snapp_prep/master/snapp_prep.rb (accessed on 30 September 2022) to convert the variant call format (VCF) file into SNAPP format. Using the parameter “-threads”, the additional SNAPP package BEAST2 was used to construct the divergence time estimation tree according to the Bayesian principle; the graphs of the results were displayed using the windows of Tracer and BEAST2 [39]. The differentiation time (2.8−4.42 Mya) of *A. albofusca* and *A. grevilleoides* was obtained [40,41]. The Bayesian tree generated in the above process was imported into Reconstruct Ancestral State in Phylogenies (RASP) [42], and the ancestral distribution area was reconstructed with the statistical dispersal-vicariance analysis (S-DIVA) method [43].

### 3.6. Population Demographic History

Stairway Plot v2 is a nonparametric or flexible model method that can be applied to low-depth sequencing and restriction site-associated DNA sequencing (RADseq) [44,45,46]. After filtering the obtained SNP data, an SNP data set with no deletions or LD was generated. The VCF file containing the SNP dataset was used to create a one-dimensional site frequency spectrum (SFS) with the python script easySFS (https://github.com/isaacovercast/easySFS (accessed on 13 October 2022)). The SFS status was designated as folded to count the SFS of minor alleles. The maximum possible number of projection values was selected to output the SFS information, and the SFS information was input into the blueprint file required for Stairway Plot to run. A total of 200 bootstraps were used to calculate the median Ne and 95% confidence intervals. The effective size of the Ne variation of the *A. grevilleoides* population was determined.

## 4. Discussion

### 4.1. Genetic Diversity of Aleuritopteris Grevilleoides

Genetic diversity assessment is one of the most effective means to understand the survival of current populations and threatened or narrowly distributed species generally have low levels of genetic diversity [47]. In this study, SNP molecular markers revealed that *A. grevilleoides* had a low level of genetic diversity (Obs_Het = 0.166, Exp_Het = 0.322, Pi = 0.333, Table 2), similar to the threatened species *Adiantum nelumboides* X.C.Zhang, *Ectopistes migratorius* L., and *Puma concolor* L. [48,49,50]. Both inbreeding and drift can reduce genetic diversity [51]. Notably, the high inbreeding coefficient of *A. grevilleoides*, Fis = 0.45, far exceeds the normal level for natural species and threatened species [48]. Population genetics theory and data from previous studies suggest that a reduction in Ne (Figure 4) leads to a loss of genetic diversity, and species with declining populations have lower genetic diversity than species with increasing or stable populations [50,51,52]. The effective population historical dynamics predicted by Ne (Figure 4) showed that the population size continued to decline and the genetic diversity of *A. grevilleoides* continued to contract; these results were consistent with the results of the genetic diversity analysis. The structure analysis (Figure 1) divided the four natural populations of *A. grevilleoides* into three genetic groups. The genetic diversity of each population of *A. grevilleoides* was lower than the overall genetic diversity, and the inbreeding coefficient was high, which will cause the genetic diversity of *A. grevilleoides* to decrease. The loss of genetic diversity can lead to the endangerment or even extinction of species [49,50].

According to these results, the genetic diversity level of *A. grevilleoides* reflects the characteristics of threatened species; this indicates that the threatened level of *A. grevilleoides* may need to be reassessed. The possible reason for the inaccurate assessment is misidentification. Figure 6A–C show the phenotypes corresponding to the genotypes of *A. grevilleoides*. For example, transitional morphology with *A. albofusca* (Figure 6D) was observed in the TD and SC populations. Yuan also found that many specimens identified as *A. grevilleoides* in the China Digital Herbarium and the National Specimen Platform were actually specimens of closely related species, such as *A. albofusca* [26].

### 4.2. Dry-Hot Valleys Are Refuges after A. grevilleoides Population Size Decreases

During the Quaternary Pleistocene, the dry-hot valley experienced several large-scale glacial periods beginning in the Tertiary Pleistocene. The dry and warm climatic conditions made the local environment a refuge for ancient plants and their descendants on the ancient southern continent, such as *Musella lasiocarpa* (Franch.) C.Y.Wu ex H.W.Li, *Dodonaea viscosa* subsp. *angustifolia* (L.f.) J.G.West, *Quercus cocciferoides* Hand.-Mazz., and *Quercus aquifolioides* Rehder and E.H.Wilson [11,12]. Based on ancestral geographic reconstruction and evolutionary time, *A. grevilleoides* was formed after the uplift of the Qinghai-Tibet Plateau (Figure 3) and three differentiations occurred with the formation of the second and third terraces of the dry-hot valley [28,29]. An ancestral population differentiated to form two branches of QH and SC/KM/TD, then, because of the changes in the environment caused by the terrace of the dry-hot valley, the population at SC/TD(C/D) and KM(B) became vicariance (Figure 3D). At the same time, the *A. grevilleoides* in Dayao County spread from SC (C) to the warmer TD (D). Previous studies have found that the main ecological factor affecting the distribution of *A. grevilleoides* is temperature, which is consistent with our results [26]. The dry-hot valleys provided a cradle for the survival and proliferation of *A. grevilleoides*, which has survived to this day.

Although dry-hot valleys provided the cradle for the spread of *A. grevilleoides*, historical events such as topographical changes have also led to its population decline. Yuan believed that environmental changes caused by human damage, such as the construction of hydropower stations and roads and slope hardening, may be the main reason for the endangerment of *A. grevilleoides* [26]. However, human activities began to affect the climate and ecosystems after the onset of the Anthropocene in 7000 BC [53]. Our research results show that the population history and size changes of *A. grevilleoides* are closely related to geo-historical events, rather than human disturbance. Therefore, human disturbances may have accelerated the process of large-scale degradation of flora in dry-hot valleys, but it is not the main cause. The time differentiation results (Figure 3A) indicated that the two differentiations of *A. grevilleoides* were closely related to the topography changes in dry-hot valley [29,54]. The results of the population dynamics analysis (Figure 4) showed that the population of *A. grevilleoides* increased once at 0.6−0.8 Mya. *A. grevilleoides* also experienced an increase in population which may have been related to the historical drainage capture events between the Luquan Wudongde and Jinpingzi Canyon 0.8 Mya. The river allowed the dispersal of the divergent *A. grevilleoides* populations in Dayao and Lukou Counties and the emergence of gene exchange events (Figure 1). The *A. grevilleoides* population decreased from 30 Kya to 40 Kya, which may be related to the latest topographic change in the dry-hot valley.

Combined with the analysis of population dynamics, it is evident that ecological restoration projects in dry-hot valleys may have been an effective means to prevent ecosystem degradation over the past decade. The population dynamics analysis indicates that the population of *A. grevilleoides* ceased to decline for a period after the ecological restoration project started. In addition, the results of the current study reveal issues in the assessment of the threat category of *A. grevilleoides*. Thus, this assessment must be re-evaluated, and corresponding protection plans should be formulated. The incorrect threat assessment may be due to an insufficient understanding of *A. grevilleoides* in the past, such as identification errors, insufficient survey scales, and a lack of systematic studies.

## 5. Conclusions

Degraded ecosystems dominated by grasses have emerged in dry-hot valleys. The predominant view is that human destruction is the main driver of environmental changes. The three differentiations of *A. grevilleoides* and the change in population dynamics were found to be related to the geological changes in the dry-hot valleys, according to our comprehensive analysis, which included genetic diversity analysis, temporal differentiation, and the reconstruction of the ancestral distribution area of *A. grevilleoides*. As early as 40 kya, the dry-hot valley effect began to intensify, resulting in a periodic decline in the population size and genetic diversity of *A. grevilleoides*. However, human activities began to affect the climate and ecosystems after the Anthropocene in 7000 BC. Therefore, human disturbances may have accelerated the process of large-scale degradation of flora in dry-hot valleys, but it is not the main cause of vegetation degradation in dry-hot valleys.

## Figures and Tables

**Figure 1 plants-12-01579-f001:**
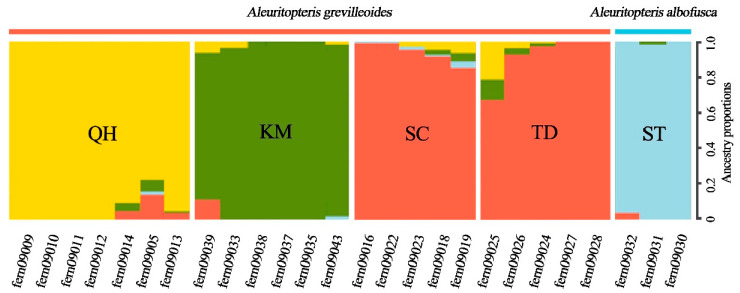
Analysis of Population Structure. Note: *Aleuritopteris grevilleoides* include QH (yellow), KM (green), SC (red), and TD (red); *A. albofusca* is ST (blue).

**Figure 2 plants-12-01579-f002:**
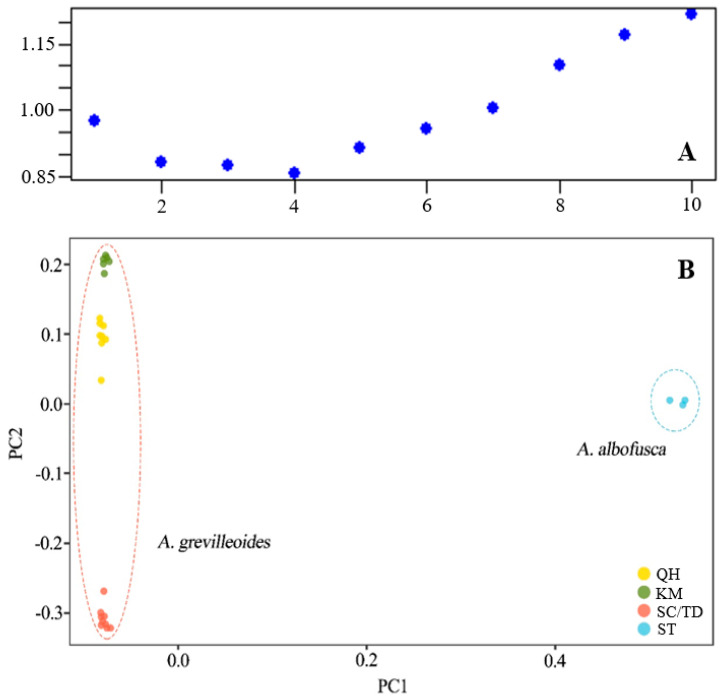
K value and PCA analysis of sampled populations: (**A**) The best grouping of K value; (**B**) The results of PCA analysis.

**Figure 3 plants-12-01579-f003:**
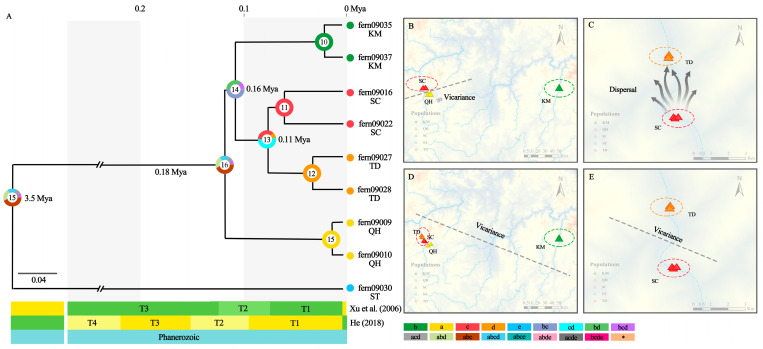
Divergence-time estimation and S-DIVA reconstruct historical biogeography: (**A**) Divergence-time estimation and S-DIVA reconstruct historical biogeography; (**B**) The node 14: The *A. grevilleoides* was probably widely distributed in the ABC area, and then a vicariance event occurred at A area; (**C**) The node 14: The original distribution area was C spread to D; (**D**) B vicariance occurred; (**E**) The node 12: a geographical separation (vicariance) between C and D has occurred. Note: The numbers inside the circle represent nodes; The terraces of dry-hot valley time node are references [28,29]; T1: The first terrace of the dry-hot valley; T2: The second terraces of the dry-hot valley; T3: The third terraces of the dry-hot valley.

**Figure 4 plants-12-01579-f004:**
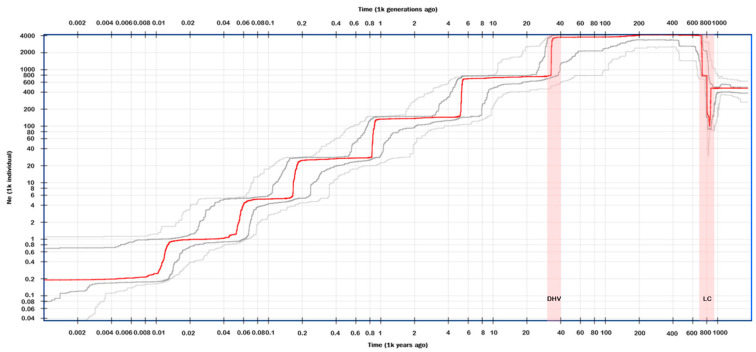
Historical fluctuation of effective population size of *A. grevilleoides*. Note: Red line: median of 200 inferences based on subsampling. Dark gray lines: 75% confidence interval of the inference. Light gray lines: 95% confidence interval of the inference. DHV: The first terraces in the dry-hot Valley; LC: the historical drainage capture events between the Luquan Wudongde and Canyon-Jinpingzi.

**Figure 5 plants-12-01579-f005:**
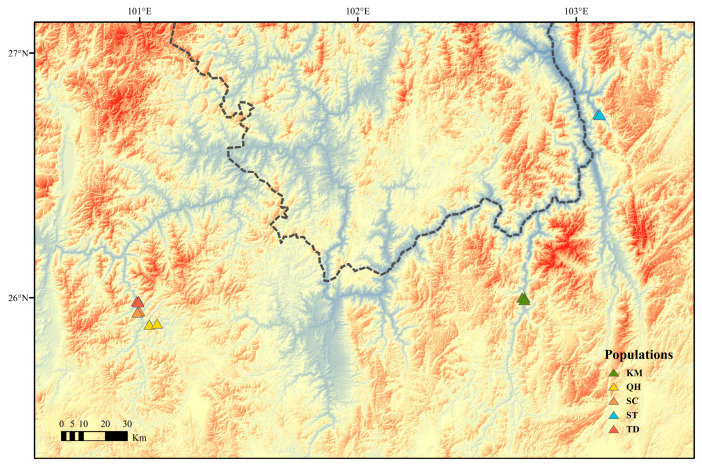
Spatial distribution points in the wild population of *Aleuritopteris grevilleoides*.

**Figure 6 plants-12-01579-f006:**
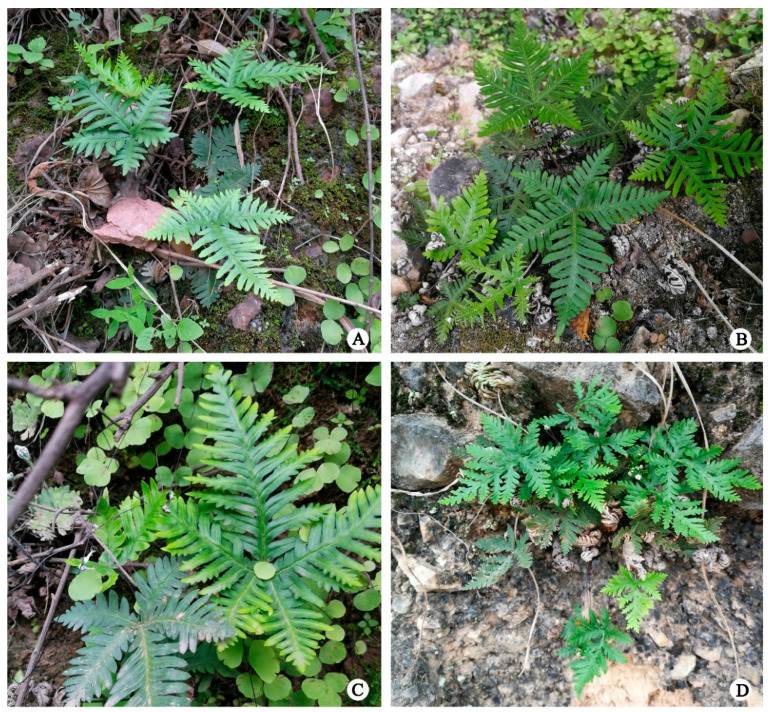
Photographs of *A. grevilleoides* and *A. albofusca* phenotypes in the wild population: (**A**) QH group (*A. grevilleoides*, fern09010); (**B**) KM group (*A. grevilleoides*, fern09033); (**C**) TD group (*A. grevilleoides*, fern09025); (**D**) ST group (*A. albofusca*, fern09032).

**Table 1 plants-12-01579-t001:** Calculation of Fst between groups of *A. grevilleoides* and *A. albofusca*.

	QH	SC	TD	KM
SC	0.225			
TD	0.188	0.031		
KM	0.226	0.142	0.116	
ST	0.684	0.621	0.543	0.579

**Table 2 plants-12-01579-t002:** Genetic diversity analysis of *A. grevilleoides*.

FID	Private	Obs_Het	Obs_Hom	Exp_Het	Exp_Hom	Pi	Fis
QH	1016	0.124	0.876	0.146	0.854	0.159	0.084
SC	439	0.182	0.818	0.222	0.778	0.251	0.144
TD	461	0.216	0.784	0.237	0.763	0.268	0.108
KM	995	0.159	0.841	0.191	0.809	0.211	0.114
*A. grevilleoides*	0	0.166	0.834	0.322	0.678	0.333	0.452

Note: Private: Private alleles; Obs_Hom: Expected homozygous; Exp_Hom: Expected homozygous; Obs_Het: Observed heterozygous; Exp_Het: Expected heterozygous; Pi: Nucleotide diversity; Fis: inbreeding coefficient.

**Table 3 plants-12-01579-t003:** Sample details of Aleuritopteris grevilleoides.

Specie	No.	Specimen Copy	Date	Site	Abbr
*A. albofusca*	fern09030	1	14 August 2019	Qiaojia County	ST
*A. albofusca*	fern09031	1	14 August 2019	Qiaojia County	ST
*A. albofusca*	fern09032	1	14 August 2019	Qiaojia County	ST
*A. grevilleoides*	fern09005	1	13 August 2019	Qihei Highway of Dayao County	QH
*A. grevilleoides*	fern09009	1	13 August 2019	Qihei Highway of Dayao County	QH
*A. grevilleoides*	fern09010	1	13 August 2019	Qihei Highway of Dayao County	QH
*A. grevilleoides*	fern09011	1	13 August 2019	Qihei Highway of Dayao County	QH
*A. grevilleoides*	fern09012	1	13 August 2019	Qihei Highway of Dayao County	QH
*A. grevilleoides*	fern09013	1	13 August 2019	Qihei Highway of Dayao County	QH
*A. grevilleoides*	fern09014	1	13 August 2019	Qihei Highway of Dayao County	QH
*A. grevilleoides*	fern09016	1	13 August 2019	SanCha River of Dayao County	SC
*A. grevilleoides*	fern09018	1	13 August 2019	SanCha River of Dayao County	SC
*A. grevilleoides*	fern09019	1	13 August 2019	SanCha River of Dayao County	SC
*A. grevilleoides*	fern09022	1	13 August 2019	SanCha River of Dayao County	SC
*A. grevilleoides*	fern09023	1	13 August 2019	SanCha River of Dayao County	SC
*A. grevilleoides*	fern09024	1	13 August 2019	Tademe Bridge of Dayao County	TD
*A. grevilleoides*	fern09025	1	13 August 2019	Tademe Bridge of Dayao County	TD
*A. grevilleoides*	fern09026	1	13 August 2019	Tademe Bridge of Dayao County	TD
*A. grevilleoides*	fern09027	1	13 August 2019	Tademe Bridge of Dayao County	TD
*A. grevilleoides*	fern09028	1	13 August 2019	Tademe Bridge of Dayao County	TD
*A. grevilleoides*	fern09033	1	17 August 2019	Luquan County of KunMing City	KM
*A. grevilleoides*	fern09035	1	17 August 2019	Luquan County of KunMing City	KM
*A. grevilleoides*	fern09037	1	17 August 2019	Luquan County of KunMing City	KM
*A. grevilleoides*	fern09038	1	17 August 2019	Luquan County of KunMing City	KM
*A. grevilleoides*	fern09039	1	17 August 2019	Luquan County of KunMing City	KM
*A. grevilleoides*	fern09043	1	17 August 2019	Luquan County of KunMing City	KM

## Data Availability

The data presented in this study are available in this article.

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
