# Peer review of "Simplified Genomic Data Revealing the Decline of Aleuritopteris grevilleoides Population Accompanied by the Uplift of Dry-Hot Valley in Yunnan, China"

_plants, 2023, doi:10.3390/plants12071579_

Round 1

Reviewer 1 Report

The MS is interesting and of interest for the Plants readership. However, Introduction needs restructuring since it gives a lot of unrelated information (not link among themselves). A bit of geo-history, irrelevant climate statements,  population information, etc. Please, try to link the statements and to introduce the reader into Your study adequately.  Also, no single word on reproduction of target species and vegetative/sexual reproduction in studied area. Please, add and explain what is known. The English in this part is hard to follow. 

The title is bombastic and needs to be changed according to Your results. I suggest Genetic structure of threaten fern Aleuritopteris grevilleoides population in Yunnan dry-hot valley, China or something similar since it reflects more those what is given in the manuscript.

Within the text You suggest geological changes to cause decline and not human impact, which is not supported by Your results. Change to fit Your concluding remarks, that both geo-changes and human impact influence decrease of population of target species.

The reference in the text and list chapter do not follow the style of the journal. Please, change.

line 47-49. In which area of the world? Please, specify.

line 50. You cannot use: dynamic changes of climatic conditions. Have You referred to significant changes of climate?

Lines, 52, 53, 55, 88, 89, 314, 315, 347, 348 etc... please insert species authorities there where they appear firstly in the manuscript. Please, check the whole manuscript.

lines 71-72 surrounding used 2 times in the same sentence. Please, reformulate

line 188. Stage et al. missing year. Anyway format of the reference should be change throughout MS.

Do not use the term endangered since it is IUCN category. Please, use instead threaten or threatened. E.g. for level in line 332, species in line 318, 364 etc.

Please explain in details what it is in Fig 7. under the figure not in the text only. The scale bar in the photos would be helpful.

lines 351-354. hard to understand. I could not. Please, reformulate!

line 365. do not use geographical changes/event but geo-historical or geological event/changes like You did in concluding remarks.

line 367. geographical changes should be relief changes or topography changes? Please, specify what are You referring to?

line 373. gene exchange should be gene exchange event(s)

line 381. degree of endangerment should be threat category

line 388. assessment of endangerment should be threat assessment

Reformulate or omit lines 393-394 from conclusions, since You cannot state that since 40Kya there was continuous population decline. This is for such a long period just Your assumption, not concluding remark based on Your results! in IUCN web You can find definition of population decline.

I am not even sure You can use this formulation for genetic diversity decline for such a long period. Cryptogams are documented to have lower genetic diversity anyway as compared to vascular plants! You give no information if the plants are reproducing nowadays asexually or sexually or both! This is a big odd of rather good set of information You have offered in this manuscript.

Author Response

Dear Reviewer,

We sincerely thank the reviewer for thoroughly examining our manuscript and providing very helpful comments to guide our revision. We have revised the manuscript accordingly, and our point-by-point responses are presented attachment.
Best regards,

Xue-Ying Wei

Reviewer 2 Report

The study describes the generation of gbs data from populations of an endangered fern species from dry-hot valleys in Yunnan, China. The SNP data are used to infer the population history through climatic changes associated with the uplift of the QuinghaiTibet plateau, an identify causes for genetic bottlenecks resulting from geological and / human related causes.

While it is an interesting study in trying to combine  data from population genetics, geological events, and habitat infringements from human activities, in my opinion the numbers of individuals per population (7/6/5/5 in case of A.grevilleoides ) is too low to base such assumptions on, I would regard ~ 10 individuals per population as lower threshold.

Aside from this, the manuscript would need some work until ready for publication. I go over some of the problems, some are concerning the language, others the presentation:

P1L29 endangered reasons and scientific protection

I assume  this means "reasons for endangerment and scientifically sound measures for protection"

P1L36 the effective population dynamics of A. grevilleoides began to contract continuously

the effective population sizes of A. grevilleoides began  to contract continuously (?)

L36 due to the dry–hot valleys of latest topographic change

sentence ! due to topographic changes resulting the generation of dry–hot valleys (?)

L37 caused by natural history

very vague, be more specific

L52 , it matrilineal diverge into  highly structured geographical units

is not a sentence

L60 Due to the uplift of the plateau, the descent of the river valley, and  the geographical effect of "foehn", the special ecological landform of "dry-hot valley" was  born

as this is central to the argumentation of this paper, one or two sentences of explanation would be fitting

L145 the high-quality DNA was constructed and sequenced

meaning "the high-quality DNA was used to construct sequencing libraries for the gbs method "(?)

L157  reads obtained by off-machine sequencing

? this is HiSeq Illumina sequencing , what means off-machine here?

L175 structure of the 26 populations

only 4 populations are investigated

L233 . The SC and TD groups had a Fst of only 0.03 and had the same genotype

then they would be identical

L240 Fig 2. NJ tree, Analysis of Population Structure and Habitat photographs of four groups.

I don't see the NJ tree in Fig 2, and there is no habitat photograph

L340 Figure 7. Schematic diagram of distribution points in the wild population.

Fig 7 shows photographs of A.grevilleoides (as correctly referenced in the text at L334, but not in the figure legend

L353 the population at QH (A) and KM(B) became vicariance (Fig. 5F) 

letters in Fig 5 only go until E, not F

For the discussion, inclusion of a graph with timelines of known geological events leading to generation of / changes in the dry hot valleys would be helpful

Author Response

Dear Reviewer,

We sincerely thank the reviewer for thoroughly examining our manuscript and providing very helpful comments to guide our revision. We have revised the manuscript accordingly, and our point-by-point responses are presented attachment.

In addition, as for the problem that the language needs to be modified properly, we have found native English speakers to help us modify it, which will take about a week. This manuscript is a revised version for the comments of reviewers.

Best regards,

Xue-Ying Wei

Reviewer 3 Report

The manuscript deals with the dry-hot valleys which formed degraded ecosystems dominated by grasses. It was viewed that human destruction is the main cause of environmental changes. Through comprehensive analysis and also genetic diversity analysis, the authors, the authors have found that the three differentiations of Aleuritopteris grevilleoides (a fern) and the change population dynamics are related to geological changes. They noticed that the human activities affected the climate and ecosystems after the Anthropocene in 7000 BC. It is concluded that the human disturbances might have accelerated the degradation of flora in dry-hot valleys.

It is a well written manuscript and deserves publication. The methods followed are adequate to draw conclusions. The objectives are clear cut and the conclusions are rational. 

Author Response

Dear Reviewer,

We sincerely thank the reviewer for thoroughly examining our manuscript and giving a high evaluation. This is the best recognition of our work in the past year, and we will work harder to do this job perfectly.

Best regards,

Xue-Ying Wei

Reviewer 4 Report

The manuscript “Simplified genomic data revealing the decline of Aleuritopteris  grevilleoides population accompanied by the uplift of the dry-hot valley in Yunnan, China” deals with the use of genomic data to understand the pattern of Aleuritopteris growth and its distribution in Yunnan valley of China. The manuscript is well-written and compiled with lots of data. I am impressed with the author's approach to taking the plant, which is very much crucial for the ecosystem.

Comments

·       The title of the manuscript needs to be revisited. Avoid word revealing.

·       Ln 51-52: Please rewrite the line.

·       Ln 58: Please provide the area in terms of latitude and longitude range.

·       Ln 103: Please provide a reference

·       Figure 1 may be written as “Spatial distribution points in the wild population of Aleuritopteris grevilleoides

·       The material and method section is well written.

·       Kindly write Fig. as Figure

·       Figures 3 and 4 can be merged as 3A and 3B

·       Figure 5 A: Please provide a good quality figure.

·       Refrain from providing references in the conclusion section. The conclusion section is the outcome of your study. Kindly provide the concluding remarks from your study.

Author Response

Refer to attachment.

Round 2

Reviewer 1 Report

The English is still hard to follow. The species names are not followed by the authorities. These needs to be improved. The MS is improved but it still needs minor changes.

Author Response

Dear Reviewer,

We sincerely thank the reviewer for thoroughly examining our manuscript and giving an evaluation. We have carefully corrected some minor errors of language and your questions. Namers are added to the species' names, and all are Latin names recognized by current authorities.

Best regards,

Xue-Ying Wei

Reviewer 2 Report

The authors apply a set of gbs generated SNPs to determine the structure of four A. grevilleoides populations. Inferred information on past vicariance events and population bottlenecks is related to the formation of dry hot valleys caused by the uplift of the Quinhai-Tibet plateau. The manuscript has been considerably improved against the first draft, but a number of problems remain, which should be adressed for clarity and readability. The main flaw of the study remains for me the low number of individuals per population.

L18 the biological evolutionary history

omit "biological"

L33 diversion of the river resulted a small amount

diversion of the river resulted in a small amount

L37 caused by geographical history might be the main reason...

eg caused by geological might be the main reason... [the history as such does not change anything, it is only our re-telling/ interpretation of past events]

L50 it genetic structure is differentiated among geographical units

sentence?

eg its genetic structure differentiated along geographical units

L51 Wu proved through research that the two increases

Wu proved that the two increases [leave research out , or be specific]

L 54 found on the Tibet Plateau of Chinese part

meaning "found on the Chinese part of the Tibet plateau" ?

L58 Dry-hot valley has a peculiar climate

Dry-hot valleys have unique climate

L59 Its formation is the result of a combination of complex geographical environment and local microclimate

this explains nothing, is this meant:

Its formation depends on a complex geographical environment and its resulting local microclimates

?

L61 reduces the humidity of the water vapor

this can't be true; is meant "reduces the water availability" ?

L  69 changed the distribution pattern of biological and natural zonality

is meant "changed the distribution of biological and geological zonality" ?

L106 vegetation of dry–hot valleys is due to the long-term interference of human activities,  which have led to land degradation and extensive soil erosion [10,28].

is meant "the degradation / endangerment of vegetation...." ?

L154 Get

determine

L216 (K) of 4 was selected, and it was divided all samples into four clusters

(K) of 4 was selected, and all samples were grouped into four clusters.

L221 /L222 In PCA1, the A. albofusca and A. grevilleoides were clearly separated

In PC1, the A. albofusca and A. grevilleoides were clearly separated [= pricipal component 1; PCA is all of the analysis with several components]

L235 there is no Figure legend for Fig 2, a short explanation of the picture is needed

L238 insert a short legend also for Fig 3

L253 The three populations of Dayao County, only the TD

Of the three 253 populations of Dayao County, only the TD [sentence!]

L263 indicated that event of vicariance may have occurred

indicated that an event of vicariance may have occurred [sentence!]

L278 there is a geographical separation 278 (vicariance) between C and D has occurred.

 a geographical separation (vicariance) between C and D has occurred.

L286 between 0.002 and 0.004 Kya

does this mean 2 to 4 years ago? similar :

L291 0.04–0.06 Kya, and 0.01–0.015

does this mean 40 to 60 / 10 to 15 years ago? If there is any information of events this recent that are relevant for this paper, it would be very important to include here (eg changes in land use, major construction projects and similar) 

L330 Fig 6 Schematic diagram..

the picture shows photographs of A.grev. phenotypes (as correctly referenced in L323 of the text), and not a schematic diagram

L340 Differentiate to form two branches...

An ancestral population differentiated to form two branches... [sentence!]

L345 is temperature, which is consistent with the results [27]

meaning "is temperature, which is consistent with our results [27]" ?

L353 / L386 after the Anthropocene in 7000 BC

after the onset of the Anthropocene in 7000 BC

L384 periodically

periodic 

Author Response

A more comprehensive modification has been made according to the requirements, as detailed in the attachment.

Reviewer 4 Report

The author made substantial changes in the manuscript. 

Author Response

Dear Reviewer,

We sincerely thank the reviewer for thoroughly examining our manuscript and giving an evaluation. We have carefully corrected some minor errors in this edition.

Best regards,

Xue-Ying Wei
